# Social–Emotional Competence among School-Aged Children in the Chinese Context: Validation of the Washoe County School District Social–Emotional Competency Assessment

**DOI:** 10.3390/bs14050399

**Published:** 2024-05-10

**Authors:** Rebecca Y. M. Cheung, Ocean O. S. Ng

**Affiliations:** 1School of Psychology and Clinical Language Sciences, University of Reading, Reading RG6 6AL, UK; 2JUST FEEL Limited, Hong Kong SAR, China; oceanng2023@justfeel.hk

**Keywords:** social–emotional competence, school-aged children, confirmatory factor analysis, Chinese context, validity

## Abstract

The present study aims to validate the 40-item and the brief 17-item Washoe County School District Social–Emotional Competency Assessment (WCSD-SECA), a self-report measure of social–emotional competencies, among Chinese school-aged children residing in Hong Kong. A total of 349 children (*M* = 9.86 years, *SD* = 1.22; 45.82% girls) and their parents (77.84% mothers) completed a set of questionnaires independently. The factor structure of both versions of the WCSD-SECA was assessed via confirmatory factor analyses. Structural equation models were then conducted to examine the predictive validity of the WCSD-SECA. The findings indicated that both the 40-item and the 17-item versions of the WCSD-SECA fit the data adequately. Both versions were also associated with self-reported positive and negative affect and parent-reported internalizing problems and externalizing problems. Additionally, social–emotional competencies as measured by the 40-item version were significantly associated with parent-reported prosocial behavior, whereas those as measured by the 17-item version were associated with prosocial behavior with a marginal significance. The findings demonstrated an adequate factor structure and predictive validity of the full version and the brief version of the WCSD-SECA in assessing social–emotional competencies. Hence, they serve as a useful tool for researchers, educators, and mental health practitioners to evaluate school-aged children’s social–emotional competencies in the Chinese context.

## 1. Introduction

Social–emotional competence lays the groundwork for child adjustment, including greater prosocial behavior, better emotional wellbeing, and fewer internalizing and externalizing problems [1,2,3]. Previous research has indicated that social–emotional learning is a fundamental process for the development of social–emotional competence [4,5]. As a result, numerous social–emotional learning programs have evolved in the school setting worldwide [6,7,8], including Hong Kong [9,10,11]. However, few cost-effective assessments have been developed that align with the definition and the specific areas of social–emotional competence, namely the skills, attitudes, and knowledge needed for demonstrating self-awareness, social awareness, self-management, relationship skills, and responsible decision making [12,13]. Even fewer assessments have been developed, adapted, or validated in the Chinese context. To provide an evidence-based tool for researchers, educators, and mental health professionals in evaluating social–emotional competence in the Chinese setting, the present study aims to validate a recently developed measure of social–emotional competence [14] in a sample of Chinese school-aged children in Hong Kong.

The Washoe County School District Social–Emotional Competency Assessment (WCSD-SECA) is a self-report instrument suitable for 5th to 12th graders [14]. Initially, the WCSD-SECA was developed through a partnership between researchers and practitioners for children in the Washoe County School District in the United States [14,15]. Following its validation, the measure has been used to assess children’s social–emotional competencies beyond the Washoe County School District within the United States [6,16,17,18]. It has also been used to evaluate social–emotional competencies among Chinese children who had participated in a school-based social–emotional learning program in rural China [19]. Unique to the instrument are five major strengths including the following: (a) a theoretical alignment with the Collaborative for Academic, Social, and Emotional Learning framework of five social–emotional learning competencies [12,13]; (b) the availability of a 40-item measure to provide a thorough assessment based on specific subscales; (c) the availability of a parallel 17-item measure in response to concerns about fatigue, time, and other needs; (d) a detailed documentation of scale development from a bank of 138 items to the 17-item and 40-item versions [14], accompanied by scale validation through focus groups and Rasch modeling, thereby demonstrating theoretical and methodological rigor [15]; (e) free accessibility for educators, researchers, and the general public. As such, the WCSD-SECA is particularly suited for use in both educational and research contexts.

Given its psychometric rigor, accessibility, and theoretical alignment with the areas of social–emotional learning [12,13], this study aims to evaluate the WCSD-SECA in a Chinese sample from Hong Kong. Importantly, the surge of social–emotional learning interventions in the Chinese context calls for valid, evidence-based assessments of social–emotional competence [9,10,11,20]. The aim of the present study is two-fold: (a) to validate the factor structure of both the 40-item and the 17-item WCSD-SECA among Chinese school-aged children and (b) to investigate the predictive validity of social–emotional competencies, as measured by the WCSD-SECA, on existing measures of positive and negative affect, internalizing problems, externalizing problems, and prosocial behavior [1,2,3]. Children’s age, gender, family income, and parents’ level of education were entered as covariates, as previous research suggested that they are associated with social–emotional competencies or other behavioral outcomes among children [21,22,23,24].

## 2. Materials and Methods

### 2.1. Procedures

Ethics approval for this study was granted by the corresponding author’s former institution (REF: 2020-2021-0310, dated 4 March 2022). Participants were recruited at primary schools through advertisements and announcements. Informed consent was obtained from children, parents, and primary schools. Among the participants, all parents and 253 children completed the questionnaires in paper-based format, whereas 96 children completed the questionnaire online due to extended social restrictions around the COVID-19 pandemic. To ensure the measure was appropriate for children with different levels of literacy, the items were read to them verbatim in a quiet classroom setting.

### 2.2. Participants

A total of 349 children (45.82% girls) and their parents were recruited from primary schools in Hong Kong. Children had a mean age of 9.86 years (*SD* = 1.22 years) and parents (77.84% mothers) had a mean age of 43.09 years (*SD* = 7.20 years). A total of 5.67% of the parents had completed primary school or below, 66.86% had completed secondary school, 16.15% had a postgraduate diploma/associate degree, 9.35% had a bachelor’s degree, 1.13% had a graduate degree, and 0.84% did not specify their education level. The median monthly household income accounting for 54.81% of the parents was HKD 10,000–30,000 (approximately USD 1282–3846), as denoted by a score of 2 on our household income scale. The median income was lower than that of the general population at HKD 34,000 (approx. USD 4359) [25]. All of the participants were ethnically Chinese.

### 2.3. Measures

*Social–emotional Competencies*. Children completed the 40-item WCSD-SECA [15,26] according to 8 subscales including the following: (a) self-awareness: self-concept; (b) self-awareness: emotion knowledge; (c) social awareness; (d) self-management: emotion regulation; (e) self-management: goal management; (f) self-management: school work; (g) relationship skills; (h) responsible decision making. To ensure the measure was appropriate for Chinese children in terms of culture and literacy levels, it was reviewed and translated from English to Chinese by the research team and an elementary school teacher following the back-translation procedures [27,28]. Children rated their ability on a scale from 1 (very difficult) to 4 (very easy). Sample items included the following: “Knowing what my strengths are” (Self-Awareness: Self-concept), “Knowing when my feelings are making it hard for me to focus” (Self-awareness: Emotion knowledge), “Knowing how my actions impact my classmates” (Social awareness), “Staying calm when I feel stressed” (Self-management: Emotion regulation), “Thinking through the steps it will take to reach my goal” (Self-management: Goal management), “Doing my schoolwork even when I do not feel like it” (Self-management: School work), “Getting along with my classmates” (Relationship skills), and “Thinking about what might happen before making a decision” (Responsible decision making). The Cronbach’s alpha of the 17-item brief version and the 40-item full version were 0.86 and 0.94, respectively. The McDonald’s omega of the 17-item brief version and the 40-item full version were 0.86 and 0.94, respectively. Table 1 shows the Cronbach’s alpha and McDonald’s omega of the subscales. Table 2 shows the specific items of each subscale.

*Positive and Negative Affect*. Children completed the Chinese version of the International Positive and Negative Affect Schedule Short Form (I-PANAS-SF) [29] to assess their positive and negative affect. The 9-item measure was previously validated in Chinese adolescents [30]. Children rated their frequency of having positive affect (e.g., inspired, determined, attentive) and negative affect (e.g., hostile, upset, ashamed) on a scale from 1 (not at all) to 5 (always). Cronbach’s alpha and McDonald’s omega are reported in Table 1.

*Child Adjustment*. Parents completed the 25-item the Strengths and Difficulties Questionnaire (SDQ) [31] to assess their children’s internalizing and externalizing problems, and prosocial behavior on a scale from 0 (not true) to 2 (certainly true). Sample items included, “Often unhappy, depressed or tearful” (internalizing problems), “Often lies or cheats” (externalizing problems), and “Often offers to help others (parents, teachers, other children)” (prosocial behavior). SDQ was previously validated in a Chinese sample [32]. Cronbach’s alpha and McDonald’s omega are reported in Table 1.

*Demographic Information*. Parents provided demographic information including children’s age, gender, parents’ education level, and monthly household income.

### 2.4. Analytic Strategy

First of all, t-tests were conducted to examine whether the children who completed the questionnaire in paper-based format differed from those who completed the questionnaire online across the variables under study. Next, zero-order correlations, means, and standard deviations were conducted as preliminary analyses. More specifically, rank-biserial *r*_pb_ was conducted between gender (i.e., a nominal binary variable) and the variables under study, whereas Kendall’s τ*_b_* was conducted between the ordinal variables under study.

Separate confirmatory factor analyses were conducted to evaluate the factor structures of the 40-item full version and 17-item brief version of the WCSD-SECA, respectively. For both versions, the raw scores were loaded on the first-order factors involving WCSD-SECA subscales. The first-order factors were then loaded on the second-order factor of social–emotional competencies.

The predictive validity of social–emotional competencies, as measured by the full version and the brief version of WCSD-SECA, on affect and child adjustment was also investigated. Specifically, structural equation models were conducted, with manifest variables of positive and negative affect, prosocial behavior, and internalizing and externalizing problems regressed on social–emotional competencies, as measured by two different versions of the WCSD-SECA. As a latent variable, social–emotional competencies were indicated by eight manifest variables of WCSD-SECA subscale scores. Children’s age, gender, family income, and parents’ level of education were entered as covariates. Maximum likelihood method was used to examine the model fit to the observed matrices of variance and covariance. 

Little’s missing completely at random test was conducted to test the null hypothesis of data missing completely at random. The finding based on the measures under study was nonsignificant, *χ*^2^(84) = 102.28, *p* = 0.09, suggesting the data were missing completely at random. However, item-level analysis did demonstrate significance, *χ*^2^(12,226) = 12,974.57, *p* < 0.001, suggesting the data at the item level were not missing completely at random. We utilized full information maximum likelihood estimation to handle the missing data. 

MPLUS, Version 8.3 [33], was used to conduct confirmatory factor analyses and structural equation models. The model fit of confirmatory factor analyses and structural equation models was evaluated via several fit indices [34]. Notably, *χ*^2^, the respective degrees of freedom, and the *p*-values were assessed, with *p* > 0.05 indicating a good model fit to the data. With a sufficiently large sample, however, the *p*-values are likely to be significant. Hence, other model fit indices were also examined, namely the Comparative Fit Index (CFI) and Tucker–Lewis index (TLI), where >0.95 indicated a good fit, whereas 0.90–0.95 indicated a reasonable fit. In addition to the CFI and the TLI, the Root Mean Square Error of Approximation (RMSEA) and the Standardized Root Mean Residual (SRMR) at <0.05 suggested a good fit, whereas 0.05–0.08 suggested a reasonable fit.

## 3. Results

Independent samples *t*-tests showed that children who completed the questionnaire in paper-based format had a higher score of self-awareness: self-concept (*M*_paper-based_ = 3.05, *SD*_paper-based_ = 0.51) than did those who completed it online (*M*_online_ = 2.87, *SD*_online_ = 0.72), *t*(347) = 2.49, *p* < 0.05. They also had greater scores of self-awareness: emotion knowledge (*M*_paper-based_ = 2.97, *SD*_paper-based_ = 0.52; *M*_online_ = 2.82, *SD*_online_ = 0.71), *t*(347) = 2.26, *p* < 0.05 and social awareness (*M*_paper-based_ = 3.00, *SD*_paper-based_ = 0.54; *M*_online_ = 2.76, *SD*_online_ = 0.68), *t*(347) = 3.48, *p* = 0.001. However, they did not differ in the rest of the social–emotional competencies, positive and negative affect, prosocial behavior, and internalizing and externalizing problems, *p*s < 0.05.

Table 1 shows the zero-order correlations, means, and standard deviations of the variables. Notably, social–emotional competencies, as indicated by the subscales of WCSD-SECA, were correlated at *p*s < 0.001. As for the demographics, children’s age was related to lower self-awareness: self-concept assessed by the 40-item WCSD-SECA (τ*_b_* = −0.10, *p* = 0.02) and the 17-item WCSD-SECA (τ*_b_* = −0.11, *p* = 0.01), as well as fewer externalizing problems (τ*_b_* = −0.09, *p* = 0.03). Being a girl was related to a lower level of parents’ education (*r*_pb_ = −0.15, *p* = 0.01), better relationship skills assessed by the 17-item WCSD-SECA (*r*_pb_ = 0.11, *p* = 0.046), and more prosocial behavior (*r*_pb_ = 0.17, *p* = 0.001). Parents’ education was also related to higher monthly household income (τ*_b_* = 0.36, *p* < 0.001) and greater self-management: goal management assessed by the 17-item WCSD-SECA (τ*_b_* = 0.11, *p* = 0.02). Finally, monthly household income was related to better self-management: goal management assessed by the 40-item WCSD-SECA (τ*_b_* = 0.11, *p* = 0.02) and the 17-item WCSD-SECA (τ*_b_* = 0.15, *p* = 0.001), fewer internalizing problems (τ*_b_* = −0.12, *p* = 0.01) and fewer externalizing problems (τ*_b_* = −0.09, *p* = 0.04). Table 2 shows the item-level descriptive statistics, including the means, standard deviations, skewness, and kurtosis of the WCSD-SECA items. Based on the data, the minimum and maximum scores of the WCSD-SECA items were 1 and 4, respectively.

A confirmatory factor analysis of the 40-item version of WCSD-SECA fit the data adequately: *χ*^2^(715) = 1142.72, *p* < 0.001, CFI = 0.90, TLI = 0.90, RMSEA = 0.04, SRMR = 0.05. Similarly, a confirmatory factor analysis of the 17-item brief version fit the data adequately: *χ*^2^(110) = 218.53, *p* < 0.001, CFI = 0.92, TLI = 0.90, RMSEA = 0.05, SRMR = 0.05 (see Table 2). For both models, all of the loadings of the first-order factors of WCSD-SECA subscales and the second-order factor of social–emotional competencies were significant at *p*s < 0.001. 

Two follow up structural equation models were conducted to examine the predictive validity of social–emotional competencies, as measured by the WCSD-SECA 40-item full version and the 17-item brief version, on positive and negative affect and child adjustment (see Figure 1 and Table 3). The model with the WCSD-SECA full version fit the data adequately: *χ*^2^(1068) = 1480.94, *p* < 0.001, CFI = 0.91, TLI = 0.90, RMSEA = 0.04, SRMR = 0.05. Specifically, social–emotional competencies as measured by the WCSD-SECA full version significantly predicted positive affect (β = 0.37, *p* < 0.001), negative affect (β = −0.15, *p* = 0.007), prosocial behavior (β = 0.13, *p* = 0.03), internalizing problems (β = −0.22, *p* < 0.001), and externalizing problems (β = −0.17, *p* < 0.003), after controlling for children’s age, gender, family income, and parents’ level of education. 

The model with the 17-item WCSD-SECA also fit the data adequately: *χ*^2^(261) = 393.84, *p* < 0.001, CFI = 0.91, TLI = 0.90, RMSEA = 0.04, SRMR = 0.05. Specifically, social–emotional competencies as measured by the brief measure significantly predicted positive affect (β = 0.38, *p* < 0.001), negative affect (β = −0.13, *p* = 0.03), internalizing problems (β = −0.23, *p* < 0.001), and externalizing problems (β = −0.16, *p* = 0.005), after controlling for children’s age, gender, family income, and parents’ level of education. The brief measure also marginally predicted prosocial behavior (β = 0.12, *p* = 0.05).

## 4. Discussion

This study evidenced the WCSD-SECA to be an adequate measure in assessing social–emotional competencies in Chinese school-aged children. The study is particularly important, given the lack of validated measures available for researchers, educators, and mental health practitioners and a surge of social–emotional learning interventions [9,10,11,20]. Through confirmatory factor analyses, both the 40-item and the 17-item WCSD-SECA demonstrated similar factor structures of social–emotional competencies involving eight factors, including the following: (a) self-awareness: self-concept; (b) self-awareness: emotion knowledge; (c) social awareness; (d) self-management: emotion regulation; (e) self-management: goal management; (f) self-management: school work; (g) relationship skills; (h) responsible decision making. The findings are comparable to the original validation studies conducted in the United States [15]. Both the full and the brief versions demonstrated predictive validity via the structural equation models, thereby suggesting that the self-report instrument is adequate in assessing social–emotional competencies in the Chinese context, particularly for school-aged children in Hong Kong.

While the 40-item and the 17-item WCSD-SECA demonstrated a reasonable fit with similar factor structures, the item “Knowing when my feelings are making it hard for me to focus” had a factor loading at 0.38 for the 17-item WCSD-SECA. The relatively low factor loading might have been due to item reduction from six to three items within the “self-awareness: emotion knowledge” subscale. Hence, future findings based on the subscale to which the item belongs should be interpreted with caution. As another caveat, based on the original studies and sources [14,26], the subscale “self-awareness: self-concept” had a single item (i.e., “Knowing what my strengths are”) in the 17-item brief WCSD-SECA. In the literature, single-item measures are not uncommon [35] given their strengths including short administration time and low data processing costs. Yet, in the present study, the subscale had a lower factor loading (at 0.44) on social–emotional competencies than did the other subscales (at 0.81–0.99). As such, rather than relying on the one-item “self-awareness: self-concept” subscale of the 17-item WCSD-SECA, the four-item subscale of the 40-item WCSD-SECA may be a more suitable to capture self-concept.

Consistent with previous studies showing the link between social–emotional competence and children’s emotional and behavioral adjustment [1], social–emotional competencies as measured by the 40-item WCSD-SECA were associated with child adjustment including greater self-reported positive affect and parent-reported prosocial behavior, and lower self-reported negative affect and parent-reported internalizing problems and externalizing problems. Social–emotional competencies as measured by the 17-item brief version was significantly associated with child adjustment outcomes, including positive and negative affect and behavioral problems. It was also associated with prosocial behavior with a marginal significance, after controlling for children’s age, gender, family income, and parents’ level of education. Based on the present findings, the brief and the full versions of WCSD-SECA performed similarly as a measure of social–emotional competencies. However, the full version had higher overall factor loadings and might be more sensitive than the brief version in statistically predicting positive behavioral outcomes, such as prosocial behavior.

In this study, girls demonstrated greater prosocial behavior, whereas a higher family income was associated with fewer internalizing problems across the zero-order correlations and structural equation models. These findings were consistent with previous research conducted in diverse contexts, including Hong Kong [36,37]. As such, future studies should take account of demographic variables as covariates in replicating the present findings.

### Strengths, Limitations, and Future Directions

The present study has strengths including validations of both the long and brief versions of WCSD-SECA in the Chinese context. We also included child and parent reports to minimize self-report bias in this study. Despite its strengths, several limitations should be noted.

First of all, the cross-sectional data precluded us from testing the directionality of effects. Future studies should use longitudinal designs and other approaches (e.g., interventions, experiments) to elucidate the precursors and consequences of social–emotional competence. Second, some of the scales had low internal consistency, with Cronbach’s alphas and McDonald’s omegas of <0.60 (e.g., positive affect, externalizing problems). Given that some of the subscales in the 17-item WCSD-SECA had one or two items, or had a very low item covariance, we were unable to generate the Cronbach’s alphas and McDonald’s omegas for several subscales. Thus, the findings, especially for the 17-item WCSD-SECA, should be interpreted with caution. In addition, the small sample precluded us from examining potential moderators. As a follow-up, future research should consider potential differences in the factor structure as a function of other factors, such as gender, age, culture, ethnicity, geographic region, and developmental period.

Consistent with validation studies of similar nature [32,38], we validated WCSD-SECA based on its original items and did not include additional items that may be culturally relevant. Even though we sought the help of experts and a local practitioner to review and translate the WCSD-SECA, additional items may be needed to examine social–emotional competencies in Hong Kong or other Chinese contexts. Hence, qualitative studies should be conducted to facilitate the inclusion of culturally relevant items. Likewise, although confirmatory factor analyses demonstrated adequate model fit, there are innumerable models that could be fit to the data. Just because the findings converged with previous studies does not mean that better fitting models cannot be found. As such, future studies should test and explore competing models and consider other factors, such as culture, developmental periods, and socioeconomic backgrounds, for item generation and scale validation. Relatedly, the zero-order correlations indicated that demographic variables such as children’s gender and parents’ education level were linked to each other. They were also more strongly linked to the core variables in the zero-order correlations than in the structural equation model. While this might have been a statistical artefact, future studies are required to further examine the links between demographics and social–emotional competencies.

Next, the median monthly household income in this study involved a wide range of HKD 10,000–30,000 (approximately USD 1282–3846). Future studies should utilize a more refined range of household income for statistical analysis. Furthermore, as some of the school-aged children were younger (around 9 years old), the items were read to all participants to standardize the procedures and rule out the potential effects of literacy levels. Given that trained researchers and teachers were needed to facilitate the completion of the self-report, the administration costs were higher than expected. Finally, the present study involved only questionnaire measures. To minimize biases, future work should utilize other forms of assessments, such as observational measures or physiological measures.

## 5. Conclusions

In conclusion, this study demonstrated adequate factor structures of the 40-item and the 17-item WCSD-SECA and their respective predictive validity in a sample of Chinese school-aged children in Hong Kong. These findings pointed to the suitability of the WCSD-SECA as a useful tool in measuring Chinese children’s social–emotional competencies. By offering an evidence-based assessment, the validated measure can serve to evaluate future SEL programs for school-aged children in the Chinese context.

## Figures and Tables

**Figure 1 behavsci-14-00399-f001:**
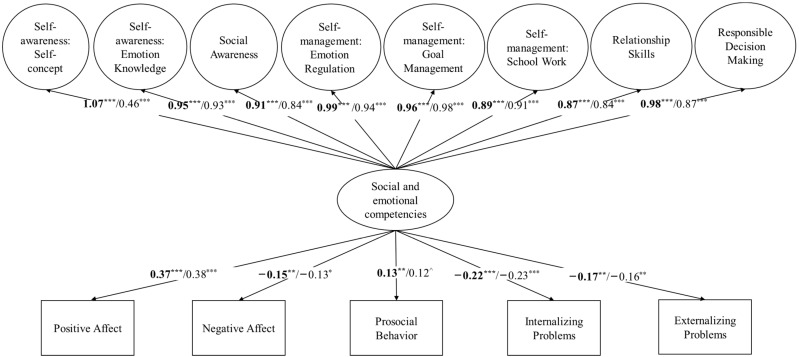
Social–emotional competencies as a predictor of children’s affect, prosocial behavior, and internalizing and externalizing problems. Estimates for the 40-item WCSD-SECA are shown in bold. The 17-item WCSD-SECA are shown in plain. Children’s age, children’s gender, parents’ education level, and monthly household income were entered as covariates for the criterion variables. ^ *p* = 0.05, * *p* < 0.05, ** *p* < 0.01, *** *p* < 0.001.

**Table 1 behavsci-14-00399-t001:** Zero-order correlations, means, and standard deviations of the variables under study.

Variable	(1)	(2)	(3)	(4)	(5)	(6)	(7)	(8)	(9)	(10)	(11)	(12)	(13)	(14)	(15)	(16)	(17)
(1) Children’s age	-	0.01	−0.05	−0.04	−0.11 *	0.02	0.01	−0.03	−0.11 *	−0.05	−0.01	0.04	−0.02	0.01	0.02	0.06	−0.09 *
(2) Children’s gender (0 = boy, 1 = girl)	**0.01**	-	−0.15 **	−0.06	0.01	0.07	0.03	0.02	0.03	0.06	0.11 *	0.08	−0.05	0.02	0.17 **	0.01	−0.09
(3) Parents’ education level	**−0.05**	**−0.15 ****	-	0.36 ***	0.08	0.05	0.07	−0.003	0.11 *	0.03	0.02	−0.03	−0.001	−0.06	0.03	−0.05	−0.002
(4) Monthly household income	**−0.04**	**−0.06**	**0.36 *****	-	0.09	0.06	0.08	0.05	0.15 **	0.09	0.02	0.04	−0.001	−0.07	0.07	−0.12 **	−0.09 *
(5) Self-awareness: self-concept	**−0.10 ***	**0.07**	**0.08**	**0.06**	-	0.38 ***	0.27 ***	0.22 **	0.30 ***	0.25 ***	0.19 ***	0.23 ***	0.24 ***	−0.12 **	0.03	−0.16 ***	−0.06
(6) Self-awareness: emotion knowledge	**−0.01**	**0.04**	**0.05**	**0.03**	**0.51 *****	-	0.39 ***	0.36 ***	0.34 ***	0.30 ***	0.29 ***	0.38 ***	0.19 ***	−0.11 **	0.04	−0.15 ***	−0.11 **
(7) Social awareness	**−0.01**	**0.04**	**0.07**	**0.06**	**0.48 *****	**0.47 *****	-	0.28 ***	0.31 ***	0.30 ***	0.33 ***	0.29 ***	0.18 ***	−0.05	0.04	−0.08	−0.03
(8) Self-management: emotion regulation	**−0.06**	**0.01**	**0.02**	**0.04**	**0.47 *****	**0.43 *****	**0.40 *****	-	0.38 ***	0.33 ***	0.30 ***	0.37 ***	0.16 ***	−0.11 **	0.03	−0.16 **	−0.08
(9) Self-management: goal management	**−0.08**	**0.06**	**0.08**	**0.11 ***	**0.15 *****	**0.42 *****	**0.41 *****	**0.49 *****	-	0.45 ***	0.30 ***	0.39 ***	0.16 ***	−0.04	0.11 *	−0.13 **	−0.12 **
(10) Self-management: school work	**−0.02**	**0.08**	**0.05**	**0.08**	**0.48 *****	**0.40 *****	**0.39 *****	**0.44 *****	**0.47 *****	-	0.36 ***	0.35 ***	0.14 **	−0.12 **	0.09 *	−0.08	−0.09 *
(11) Relationship skills	**−0.07**	**0.04**	**0.09 ***	**0.03**	**0.41 *****	**0.40 *****	**0.43 *****	**0.38 *****	**0.36 *****	**0.38 *****	-	0.28 ***	0.21 ***	−0.09 *	0.02	−0.09 *	−0.07
(12) Responsible decision making	**0.01**	**0.04**	**0.02**	**0.01**	**0.51 *****	**0.46 *****	**0.45 *****	**0.49 *****	**0.51 *****	**0.49 *****	**0.40 *****	-	0.16 ***	−0.08	0.07	−0.11 **	−0.14 ***
(13) Positive affect	**−0.02**	**−0.05**	**−0.001**	**−0.001**	**0.26 *****	**0.21 *****	**0.23 *****	**0.20 *****	**0.17 *****	**0.18 *****	**0.25 *****	**0.20 *****	-	0.18 ***	0.02	−0.12 **	−0.01
(14) Negative affect	**0.01**	**0.02**	**−0.06**	**−0.07**	**−0.13 *****	**−0.12 ****	**−0.05**	**−0.08 ***	**−0.05**	**−0.11 ****	**−0.90 ***	**−0.10 ***	**0.18 *****	-	−0.06	0.15 ***	0.13 ***
(15) Prosocial behavior	**0.002**	**0.17 ****	**0.03**	**0.07**	**0.08**	**0.09 ***	**0.07**	**0.06**	**0.09 ***	**0.10 ***	**0.08**	**0.07**	**0.02**	**−0.06**	-	−0.20 ***	−0.23 ***
(16) Internalizing problems	**0.06**	**0.01**	**−0.05**	**−0.12 ****	**−0.16 *****	**−0.15 *****	**−0.13 *****	**−0.16 *****	**−0.14 *****	**−0.14 *****	**−0.13 ****	**−0.14 *****	**−0.12 ****	**0.15 *****	**−0.20 *****	-	0.39 ***
(17) Externalizing problems	**−0.09 ***	**−0.09**	**−0.002**	**−0.09 ***	**−0.10 ***	**−0.10 ***	**−0.07**	**−0.10 ***	**−0.11 ****	**−0.14 *****	**−0.06**	**−0.12 ****	**−0.01**	**0.13 *****	**−0.23 *****	**0.39 *****	-
***M* (40-item measure)**	**9.86**	-	**2.33**	**2.57**	**3.00**	**2.93**	**2.93**	**2.76**	**2.80**	**3.04**	**2.96**	**2.95**	**12.14**	**10.68**	**6.90**	**4.86**	**5.66**
***SD* (40-item measure)**	**1.21**	-	**0.77**	**1.04**	**0.58**	**0.58**	**0.59**	**0.63**	**0.68**	**0.59**	**0.57**	**0.60**	**3.40**	**3.74**	**2.08**	**3.22**	**2.79**
**Cronbach’s alpha (40-item measure)**	-	-	-	-	**0.59**	**0.68**	**0.68**	**0.60**	**0.77**	**0.78**	**0.69**	**0.69**	**0.55**	**0.75**	**0.77**	**0.70**	**0.58**
**McDonald’s omega (40-item measure)**	-	-	-	-	**0.59**	**0.68**	**0.68**	**0.60**	**0.78**	**0.78**	**0.68**	**0.69**	**0.54**	**0.74**	**0.77**	**0.70**	**0.64 ^§^**
*M* (17-item measure)	9.86	-	2.33	2.57	2.85	2.90	2.91	2.83	2.79	3.19	3.22	2.99	12.14	10.68	6.90	4.86	5.66
*SD* (17-item measure)	1.21	-	0.77	1.04	0.86	0.62	0.66	0.73	0.71	0.61	0.59	0.69	3.40	3.74	2.08	3.22	2.79
Cronbach’s alpha (17-item measure)	-	-	-	-	NA ^‡^	0.48	0.57	0.40	0.53	0.59	0.57	0.52	0.55	0.75	0.77	0.70	0.58
McDonald’s omega (17-item measure)	-	-	-	-	NA ^‡^	0.51	0.58	NA ^∆^	NA ^∆^	NA ^∆^	NA ^∆^	NA ^∆^	0.54	0.74	0.77	0.70	0.64 ^§^

Note. * *p <* 0.05, ** *p* < 0.01, *** *p* < 0.001. Variables (5)–(12) are subscales of the WCSD-SECA. Estimates for the 40-item WCSD-SECA are shown in bold, and those of the 17-item WCSD-SECA are shown in plain. Monthly household income scale ranging from 1 to 6: 1 = less than or equal to HKD 10,000; 2 = HKD 10,001–30,000; 3 = HKD 30,001–50,000; 4 = HKD 50,001–70,000; 5 = HKD 70,001–90,000; 6 = equal to or greater than HKD 90,001. Rank-biserial *r*_pb_ was conducted between gender (i.e., a nominal binary variable) and the variables under study, whereas Kendall’s τ*_b_* was conducted between the ordinal variables under study. ^‡^ Cronbach’s alpha and McDonald’s omega were not available as the subscale involved a single item. ^§^ The SDQ item “Generally obedient, usually does what adults request”. was removed in the calculation of McDonald’s omega due to zero item covariance. ^∆^ McDonald’s omega could not be estimated as the number of items of the subscale was less than 3.

**Table 2 behavsci-14-00399-t002:** Descriptive statistics, standardized factor loadings, and unstandardized factor loadings (standard errors) of the 40-item and 17-item WCSD-SECA based on confirmatory factor analyses.

Factor	Mean	*SD*	Skewness	Kurtosis	40-Item WCSD-SECA		17-Item WCSD-SECA	
					Unstandardized (*SE*)	Standardized	Unstandardized (*SE*)	Standardized
**First-order factor**								
Self-awareness: Self-concept								
→ Knowing what my strengths are ^◊^	2.85	0.86	−0.36	−0.52	1.00 (0.00) ^f^	0.41	0.16 (0.03)	0.44
→ Knowing how to get better at things that are hard for me to do at school	2.81	0.89	−0.29	−0.67	1.65 (0.22)	0.65	-	-
→ Knowing when I am wrong about something	3.27	0.69	−0.40	−0.85	1.07 (0.16)	0.54	-	-
→ Knowing when I can’t control something	3.18	0.70	−0.26	−0.97	1.07 (0.17)	0.52	-	-
Self-awareness: Emotion Knowledge								
→ Knowing when my feelings are making it hard for me to focus	2.71	0.88	−0.23	−0.64	1.00 (0.00) ^f^	0.49	1.00 (0.00) ^f^	0.38
→ Knowing the emotions I feel	3.25	0.70	−0.38	−0.91	0.87 (0.13)	0.53	1.09 (0.22)	0.52
→ Knowing ways to make myself feel better when I’m sad	2.81	0.97	−0.41	−0.79	1.28 (0.17)	0.57	-	-
→ Noticing what my body does when I am nervous	3.15	0.71	−0.23	−1.00	0.88 (0.13)	0.53	-	-
→ Knowing when my mood affects how I treat others	2.94	0.85	−0.46	−0.40	1.27 (0.16)	0.65	-	-
→ Knowing ways I calm myself down	2.83	0.95	−0.84	−0.63	1.27 (0.17)	0.58	1.65 (0.30)	0.59
Social Awareness								
→ Learning from people with different opinions than me	2.78	0.89	−0.32	−0.62	1.00 (0.00) ^f^	0.50	1.00 (0.00) ^f^	0.56
→ Knowing what people may be feeling by the look on their face	2.90	0.95	−0.47	−0.72	0.96 (0.14)	0.46	1.05 (0.15)	0.55
→ Knowing when someone needs help	3.12	0.70	−0.17	−0.95	0.97 (0.13)	0.61	0.88 (0.12)	0.62
→ Knowing how to get help when I’m having trouble with a classmate	3.02	0.85	−0.47	−0.54	1.28 (0.15)	0.68	-	-
→ Knowing how my actions impact my classmates	2.92	0.84	−0.50	−0.27	1.18 (0.15)	0.63	-	-
Self-management: Emotion Regulation								
→ Getting through something even when I feel frustrated	2.78	0.92	−0.30	−0.76	1.00 (0.00) ^f^	0.53	1.00 (0.00) ^f^	0.49
→ Being patient even when I am really excited	2.87	0.92	−0.50	−0.55	0.96 (0.13)	0.51	1.03 (0.15)	0.51
→ Staying calm when I feel stressed	2.77	0.93	−0.28	−0.79	1.04 (0.13)	0.54	-	-
→ Working on things even when I don’t like them	2.63	0.98	−0.16	−0.96	1.01 (0.13)	0.50	-	-
Self-management: Goal Management								
→ Finishing tasks even if they are hard for me	2.87	0.81	−0.26	−0.51	1.00 (0.00) ^f^	0.67	1.00 (0.00) ^f^	0.68
→ Setting goals for myself	2.70	0.92	−0.14	−0.86	0.94 (0.10)	0.55	0.89 (0.11)	0.53
→ Reaching goals that I set for myself	2.75	0.92	−0.27	−0.75	1.03 (0.10)	0.61	-	-
→ Thinking through the steps it will take to reach my goal	2.85	0.90	−0.46	−0.50	1.06 (0.10)	0.64	-	-
Self-management: School Work								
→ Doing my schoolwork even when I do not feel like it	3.25	0.70	−0.38	−0.91	1.00 (0.00) ^f^	0.67	1.00 (0.00) ^f^	0.67
→ Being prepared for tests	3.15	0.70	−0.22	−0.95	0.89 (0.10)	0.60	0.98 (0.11)	0.66
→ Working on assignments even when they are hard	3.27	0.71	−0.44	−0.94	1.02 (0.09)	0.66	-	-
→ Planning ahead so I can turn a project in on time	2.93	0.85	−0.35	−0.63	1.12 (0.12)	0.63	-	-
→ Finishing my schoolwork without reminders	3.15	0.72	−0.23	−1.03	0.96 (0.10)	0.63	-	-
→ Staying focused in class even when there are distractions	2.70	0.88	−0.08	−0.77	1.17 (0.12)	0.64	-	-
Relationship Skills								
→ Respecting a classmate’s opinions during a disagreement	3.19	0.68	−0.24	−0.85	1.00 (0.00) ^f^	0.59	1.00 (0.00) ^f^	0.63
→ Getting along with my classmates	3.28	0.71	−0.45	−0.92	1.08 (0.13)	0.60	1.04 (0.14)	0.63
→ Sharing what I am feeling with others	2.63	1.04	−0.21	−1.10	1.11 (0.17)	0.43	-	-
→ Talking to an adult when I have problems at school	2.77	1.03	−0.41	−0.98	1.29 (0.18)	0.50	-	-
→ Being welcoming to someone I don’t usually eat lunch with	2.89	0.90	−0.48	−0.52	1.06 (0.15)	0.47	-	-
→ Getting along with my teachers	3.17	0.67	−0.21	−0.79	1.08 (0.13)	0.64	-	-
Responsible Decision Making								
→ Thinking about what might happen before making a decision	2.87	0.88	−0.39	−0.56	1.00 (0.00) ^f^	0.55	1.00 (0.00) ^f^	0.62
→ Knowing what is right or wrong	3.13	0.73	−0.21	−1.08	0.85 (0.11)	0.56	0.80 (0.10)	0.60
→ Thinking of different ways to solve a problem	2.98	0.84	−0.52	−0.29	1.15 (0.12)	0.67	-	-
→ Saying “no” to a friend who wants to break the rules	3.01	0.91	−0.59	−0.50	1.00 (0.13)	0.54	-	-
→ Helping to make my school a better place	2.77	0.95	−0.31	−0.83	1.15 (0.14)	0.59	-	-
**Second order factor**								
→ Social–Emotional Competencies								
→ Self-awareness: Emotion Knowledge					1.00 (0.00) ^f^	1.06	1.00 (0.00) ^f^	0.91
→ Social Awareness					1.11 (0.18)	0.92	1.37 (0.27)	0.84
→ Self-management: Emotion Regulation					1.29 (0.19)	0.99	1.47 (0.29)	0.99
→ Self-management: Goal Management					1.38 (0.19)	0.95	1.75 (0.31)	0.99
→ Self-management: School Work					1.12 (0.16)	0.87	1.35 (0.26)	0.88
→ Relationship Skills					0.93 (0.14)	0.87	1.13 (0.22)	0.81
→ Responsible Decision Making					1.27 (0.19)	0.97	1.58 (0.30)	0.89
→ Self-awareness: Self-concept ^◊^					1.02 (0.15)	0.88	0.16 (0.03)	0.44

Note. All factor loadings were significant at *p* < 0.001. ^f^ Indicates fixed factor loading. ^◊^ As the subscale “Self-awareness: Self-concept” consisted of a single indicator in the 17-item WCSD-SECA, the factor variance was fixed to 1.

**Table 3 behavsci-14-00399-t003:** Standardized and unstandardized parameter estimates (standard errors) of the structural equation model with social–emotional competencies, as measured by the 40-item and 17-item WCSD-SECA, as predictors of child adjustment.

Parameter	40-Item WCSD-SECA		17-Item WCSD-SECA	
	*B* (*SE*)	β	*B* (*SE*)	β
Structural Model				
Social–Emotional Competencies				
→ Positive emotions	3.40 (0.64)	0.37 ***	3.95 (0.84)	0.38 ***
→ Negative emotions	−1.53 (0.60)	−0.15 **	−1.48 (0.71)	−0.13 *
→ Prosocial behavior	0.70 (0.33)	0.13 *	0.71 (0.38)	0.12 ^
→ Internalizing problems	−1.91 (0.54)	−0.22 ***	−2.19 (0.65)	−0.23 ***
→ Externalizing problems	−1.26 (0.45)	−0.17 **	−1.36 (0.53)	−0.16 **
Covariates				
Children’s age				
→ Positive emotions	−0.03 (0.15)	−0.01	0.02 (0.14)	0.01
→ Negative emotions	0.05 (0.17)	0.02	0.04 (0.16)	0.01
→ Prosocial behavior	−0.09 (0.09)	−0.05	−0.01 (0.09)	−0.01
→ Internalizing problems	0.12 (0.14)	0.04	0.15 (0.14)	0.06
→ Externalizing problems	−0.24 (0.13)	−0.10	−0.28 (0.12)	−0.12 *
Children’s gender (0 = boy, 1 = girl)				
→ Positive emotions	−0.43 (0.36)	−0.06	−0.51 (0.34)	−0.07
→ Negative emotions	−0.02 (0.41)	−0.00	0.06 (0.40)	0.01
→ Prosocial behavior	0.62 (0.23)	0.15 **	0.70 (0.22)	0.17 **
→ Internalizing problems	0.43 (0.35)	0.07	0.27 (0.25)	0.04
→ Externalizing problems	−0.39 (0.30)	−0.07	−0.47 (0.29)	−0.08
Parents’ education level				
→ Positive emotions	−0.11 (0.27)	−0.02	−0.13 (0.25)	−0.03
→ Negative emotions	0.22 (0.31)	0.05	0.08 (0.30)	0.02
→ Prosocial behavior	0.19 (0.17)	0.07	0.20 (0.17)	0.08
→ Internalizing problems	0.17 (0.26)	0.04	0.16 (0.25)	0.04
→ Externalizing problems	0.13 (0.22)	0.04	0.12 (0.22)	0.03
Monthly household income				
→ Positive emotions	−0.10 (0.21)	−0.03	−0.21 (0.19)	−0.07
→ Negative emotions	−0.52 (0.24)	−0.14	−0.39 (0.23)	−0.11
→ Prosocial behavior	−0.02 (0.13)	−0.01	0.02 (0.13)	0.01
→ Internalizing problems	−0.49 (0.20)	−0.15 *	−0.41 (0.19)	−0.13 *
→ Externalizing problems	−0.27 (0.17)	−0.10	−0.31 (0.17)	−0.12

Note. ^ *p* = 0.05, * *p <* 0.05, ** *p* < 0.01, *** *p* < 0.001. Monthly household income scale ranging from 1 to 6: 1 = less than or equal to HKD 10,000; 2 = HKD 10,001–30,000; 3 = HKD 30,001–50,000; 4 = HKD 50,001–70,000; 5 = HKD 70,001–90,000; 6 = equal to or greater than HKD 90,001.

## Data Availability

The dataset analyzed in this article is not publicly available. Requests to access the dataset should be directed to Rebecca Y. M. Cheung.

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
