# Peer review of "Social–Emotional Competence among School-Aged Children in the Chinese Context: Validation of the Washoe County School District Social–Emotional Competency Assessment"

_behavsci, 2024, doi:10.3390/bs14050399_

Round 1
Reviewer 1 Report
Comments and Suggestions for Authors
Please see the attached feedback.

Only minor issues are highlighted which are noted in the attached feedback.
Author Response
Please refer to the attached response letter. Thank you.

Reviewer 2 Report
Comments and Suggestions for Authors
Thank you for your paper. Please reconsider your CFA analysis as it seems incorrect for the short version. Overall, more should be done in order to describe the results and explain them in the discussion. The paper is short, and it is good for readers, however, it has insufficient descriptions and explanations.
Please find specific comments:
1. Please present more information regarding relevance of the studied construct in the introduction. Moreover, the construct of social-emotional competence should be described.
2. Please indicate when the study was conducted.
3. Please use zeroes before full stops in numbers (e.g., not .55 but 0.55) as this is acc. to journal's guidelines.
4. Internal consistency reliability coefficients should be presented in Table 1.
5. Please clearly indicate a type of correlations. Pearson?
6. CFA fit indices with their cut-offs should be described in the analytic section.
7. Abbreviations were used incorrectly. Please decipher abbreviations when introducing them for the first time.
8. Long paragraphs are unwanted, especially in the results section where different analyses were described in the same paragraphs. Very hardly to read.
9. Some factor loadings were low, however, this was not explained.
10. The first factor in the short measure comprised only with 1 item? Is it true? This seems incorrect. Please see requirements for the number of factors in CFA.
11. Please present M, SD, skewness, kurtosis, min and max. values for all items across two measures (for example, in Supplementary Material).
Author Response

(The authors gave the same response as above.)

Round 2
Reviewer 2 Report
Comments and Suggestions for Authors
Thanks for improvements, please see other comments how to improve the paper.
1. Some phrases should be proofread, e.g., "the 40-item version was significantly associated with parent-reported prosocial behavior, whereas the 17-item version was associated with prosocial behavior with a marginal significance". How can the 40-item scale correlate with smth? psychological constructs as measured by scales can correlate with each other.
2. Line 62: "and € open-source w". What does this "€" mean?
3. Please indicate a type of correlation used for the scale variables.
4. Zeroes before full stops in numbers should be used. The authors corrected this, however, there are a lot of place where this problem was not resolved.
5. Lines 296-336: Long paragraphs are unwanted. Please see these lines. Please divide the information into paragraphs. "Finally," in the end of line 336 seems incomplete.
6. Please indicate a date of ethical approval.
7. Please describe procedure before participants as you plan first procedure, and then invite participants acc. to the procedure.
Round 3
Reviewer 2 Report
Comments and Suggestions for Authors
Thanks for improvements. Zeroes before full stops in numbers were corrected, however, in the not all places (e.g., line 187, some places in Table 2). Hope the authors will resolve this issue with more attention.
